# Expression of Mutant Glycine Receptors in *Xenopus* Oocytes Using Canonical and Non-Canonical Amino Acids Reveals Distinct Roles of Conserved Proline Residues

**DOI:** 10.3390/membranes12101012

**Published:** 2022-10-19

**Authors:** Sarah C. R. Lummis, Dennis A. Dougherty

**Affiliations:** 1Department of Biochemistry, University of Cambridge, Tennis Court Road, Cambridge CB2 1GA, UK; 2Division of Chemistry and Chemical Engineering, California Institute of Technology, Pasadena, CA 91125, USA

**Keywords:** ligand-gated ion channel, Cys-loop receptor, cis prolyl peptide bond, noncanonical amino acid, mutagenesis, electrophysiology

## Abstract

Pentameric ligand-gated ion channels (pLGIC) play important roles in fast neuronal signal transmission. Functional receptors are pentamers, with each subunit having an extracellular domain (ECD), a transmembrane domain (TMD) and an intracellular domain. The binding of the agonist to the ECD induces a structural change that is transduced to the TMD to open the channel. Molecular details of this process are emerging, but a comprehensive understanding is still lacking. Proline (Pro) is one amino acid that has attracted much interest; its unusual features generate bends in loops and kinks and bulges in helices, which can be essential for function in some pLGICs. Here, we explore the roles of four conserved Pros in the glycine receptor (GlyR), creating substitutions with canonical and noncanonical amino acids, characterizing them using two electrode voltage clamp electrophysiology in *Xenopus* oocytes, and interpreting changes in receptor parameters using structural data from the open and closed states of the receptor. The data reveal that for efficient function, the Pro in the α1β1 loop is needed to create a turn and to be the correct size and shape to interact with nearby residues; the peptide bond of the Pro in the Cys-loop requires the *cis* conformation; and the Pros in loop A and M1 allow efficient function because of their reduced hydrogen bonding capacity. These data are broadly consistent with data from other pLGICs, and therefore likely represent the important features of these Pros in all members of the family.

## 1. Introduction

Cys-loop receptors, which include nicotinic acetylcholine (nACh), 5-hydroxytryptamine (HT)_3_, γ-aminobutyric acid (GABA)_A_, and glycine (Gly) receptors (R), are pentameric ligand-gated ion channels (pLGIC) responsible for fast excitatory and inhibitory synaptic neurotransmission in the central and peripheral nervous systems [1,2,3,4]. Members of this family are pentameric, with each of the subunits having an extracellular domain (ECD), a transmembrane domain (TMD), and an intracellular domain. Molecules that activate these receptors bind at the interface between two adjacent subunits in the ECD, triggering a conformational change that ultimately opens the ion channel in the TMD. 

Recent high-resolution structural data from both eukaryotic and prokaryotic pLGICs have begun to clarify their mechanism of action [5,6,7,8,9,10,11], but as these are only snapshots in a particular state, much still remains to be discovered. The structures do, however, allow us greater insight into the specific roles of amino acids in these important proteins, which should ultimately allow us to understand how they contribute to protein function. Gly receptors (GlyR) have been particularly well examined to date, with at least 5 high resolution structures in the database; thus, these proteins are good candidates for this type of study.

Some residues are more likely than others to play roles in receptor function, and one of these is proline (Pro), which has been identified as being important for the structure and/or function of many proteins. This is largely due to its unusual characteristics compared to the other 19 canonical amino acids: it has restricted H bonding capability, increased steric bulk at the backbone nitrogen, and a greater propensity for its backbone peptide bond to exist in a *cis* conformation—indeed a Pro with a *cis* peptide is the second most conserved residue type, the first being half cystine [12].

Pro has already been shown to be important for function in a variety of locations in pLGICs, including the M2-M3 loop, M1 and M4, with the most conserved Pro being in the eponymous Cys-loop, leading to the proposal that the family should be called Pro-loop receptors [13]. Understanding the roles of Pros is therefore essential for understanding both similarities and differences in the structures and molecular mechanism of action of these receptors.

To explore the specific roles and interactions of conserved Pros in the pLGIC, we used those in the GlyR (Figure 1). Each was initially substituted with alanine and functionally characterised, and then they were further explored using noncanonical amino acid mutagenesis with a range of Pro analogues. The data reveal which features of Pro are important at each location, and how these might contribute to the function of the receptor.

## 2. Materials and Methods

### 2.1. Molecular Biology

The cDNA for the human GlyR α1 subunit was in pGEMHE (ThermoFischer Scientific, Bishops Stortford, UK). Site-directed mutagenesis was performed using QuikChange (Agilent Technologies, Milton Keynes, UK) to generate the appropriate codon. For non-canonical amino acid mutants and conventional mutants generated by nonsense suppression, the site of interest was mutated to the TAG stop codon. Receptor mRNA was then prepared by in vitro runoff transcription using the Ambion T7 mMessage mMachine kit (ThermoFischer Scientific, Bishops Stortford, UK). Noncanonical amino acids were ligated to tRNA as previously described [14].

### 2.2. Oocyte Preparation and RNA Injection 

Stage V-VI oocytes of *Xenopus laevis* were harvested and injected with RNAs as described previously [14]. For nonsense suppression experiments, each cell was injected with 50–100 ng each of receptor mRNA and tRNA charged with the appropriate Pro analogue ~48 h before recording. For wild type and conventional mutants, each cell received a single injection of 10–25 ng of receptor mRNA ~24 h before recording. Injection volumes were 50–100 nL per cell. 

As a negative control for suppression experiments at each site, unligated tRNA was co-injected with mRNA in the same manner as charged tRNA. These experiments yielded negligible responses. Pro recovery (injecting tRNA charged with Pro to regenerate a wild type channel via nonsense suppression) was used alongside Pro analogues as a positive control.

### 2.3. Electrophysiology

Two-electrode voltage clamping of *Xenopus* oocytes was performed using standard electrophysiological procedures as previously [14,15], using an OpusXpress system (Axon Instruments, Inc., Union City, CA, USA). All experiments were performed at 22–25 °C. Gly or taurine (Tau) (Sigma Aldrich, Gillingham, UK) were diluted in ND96 (NaCl 96 mM, KCl 2 mM, CaCl_2_ 1.8 mM, MgCl_2_ 1 mM, HEPES 20 mM, pH 7.5), and delivered to cells via a computer-controlled perfusion system. Glass microelectrodes were backfilled with 3 M KCl and had a resistance of approximately 1 MΩ. The holding potential was −60 mV unless otherwise specified. Agonist-elicited currents were normalized to the maximum current in each oocyte; for Tau this was both to the Tau maximal current and the Gly maximal current as indicated. Data were analysed using Prism (GraphPad Software Inc., San Diego, CA, USA), fitting concentration-response data to the four-parameter logistic equation. Statistical analysis was performed using ANOVA with a Dunnett’s multiple comparisons post test.

### 2.4. Structures

PDBs 5VDH (open GlyR) and 5CFB (closed GlyR), were downloaded from the PDB database and viewed and or mutated using PyMol (v2.4., Pymol, Cambridge, UK) or Swiss-PDBViewer (v4.1, Basel, Switzerland). The GlyR structures are those of the homomeric α3 GlyR, but the majority of residues, and in particular all the proline residues studied here, are identical in the α1 and α3 GlyR.

## 3. Results 

The Pros examined in this study were chosen because they are conserved in the pLGIC family and had previously been shown contribute to GlyR function. Initially, each was substituted with Ala, and changes in functional characteristics were determined following expression in *Xenopus* oocytes. The data unexpectedly revealed that all of them yielded functional receptors, albeit with some changes in response kinetics and a decrease in potency (Figure 2 and Figure 3 and Table 1). 

P96A, P146A, and P230A-containing receptors all had increased EC_50_ values compared with WT. As EC_50_ = K_A_(1 + E) where E = efficiency, it is a function of both binding and gating; it is however possible to get an indication of changes in E by examining the response to a partial agonist. Therefore, to further explore these receptors, we examined the parameters of taurine activation. Taurine in our hands yielded 50% of the maximal WT glycine-induced response. In our mutant receptors, this value was significantly lower (Figure 3 and Figure 4, Table 2), suggesting that the change in EC_50_ is more likely to be caused by a change in efficacy, i.e., gating, than binding affinity. This suggestion is consistent with the location of these Pros some distance from the binding site, and with the changes in kinetics of the responses (Figure 1 and Figure 3).

These data, however, do not provide information as to which characteristics of Pro result in efficient gating; to explore this, we substituted the Pros with a range of Pro analogues using noncanonical mutagenesis and nonsense suppression. These receptors were compared to WT receptors generated using nonsense suppression (Pro recovery); EC_50_s from these receptors were slightly higher than for WT; this is likely due to lower expression levels as GlyR EC_50_s are inversely related to I_max_ [16]. The data (Figure 5 and Table 2) reveal very different patterns with the different analogues in different locations.

We also performed the same experiments on P30, whose substitution with Ala in HEK cells but not oocytes results in non-functional receptors [17]. In our hands P30A-substituted receptors had a Gly EC_50_ of 112 μM (pEC_50_ = 3.95 ± 0.02) while the value for Tau was 1230 μM (pEC_50_ = 2.91 ± 0.06); data = mean + SEM, n = 5. I_max_Tau/I_max_Gly (60 ± 5 %) was not significantly different to WT (50 ± 4%).

For P30, incorporation of *cis*-4-fluoroproline (CFP) and *trans*-4-fluoroproline (TFP), which differ from Pro only in a single H to F exchange, yielded responses similar to WT. However changing the size and shape by addition of a methyl group had different effects depending on its location: if at the 2 position (2-methylproline, 2MeP) EC_50_ was increased, while if at the 3 position (3-methylproline, 3MeP), it was decreased. Pipecolic acid (Pip), which has a larger ring than Pro, also decreased EC_50_, as did α-hydroxyvaline (Vah). Vah generates receptors with a backbone ester in place of an amide, and thus probes the importance of the decreased hydrogen bonding capacity of Pro. For P96, all the Pro analogues yielded receptors with similar or slightly increased EC_50_s compared to Pro, the exception being 3MeP, which caused a large increase in EC_50_. For P146 the opposite effects of CFP and TFP on EC_50_ suggests the *cis*-bias of the prolyl peptide bond is important: TFP (increased EC_50_) has a *trans* bias, while CFP (decreased EC_50_) has a *cis* bias. Consistent with this hypothesis 2MeP (increased EC_50_) and Pip (decreased EC_50_) have a *trans* and *cis* bias respectively. For P230 all the Pro analogues had similar EC_50_s to Pro, apart from 3MeP, which resulted in a decreased EC_50_.

The Pro at the apex of the Cys-loop (here P146) has been the most intensively explored Pro in other pLGIC, so we also examined the action of Tau on receptors containing some of the Pro analogues. The data (Figure 6, Table 3) show different I_max_Tau /I_max_Gly for each of the four selected examples, supporting our hypothesis that changes to EC_50_ are due to alteration in gating and not binding. 

## 4. Discussion

Our study, using both canonical and noncanonical mutagenesis, reveals which features of conserved GlyR Pros are important and how these might contribute to the function of the receptor. Pros have a range of unusual properties, and the specific property that is required differs between the differently located Pros, emphasizing the versatility of this unusual amino acid in contributing to protein function. Our functional data, combined with structural information from the open and closed states of the GlyR, reveal the ability of Pros to generate bends and/or turns, its enhanced *cis* bias, and/or its reduced H bonding capacity are necessary at specific locations for the efficient function of this protein. These Pros are largely conserved, and in other pLGIC some have been explored in as much detail as we do here, revealing similarities but also a number of minor differences between the different members of this family. Each Pro is discussed in more detail below.

P30 was previously shown to be an important GlyR residue as its substitution with Ala prevented function in HEK cells, but in oocytes robust currents were observed, albeit with an increased EC_50_. This was likely due to effects on binding affinity rather that gating as I_max_Tau/I_max_Gly in P30A-containing receptors was similar to WT [17]. Ala substitution of the equivalent Pro in other pLGIC has a variety of effects: in the 5-HT_3_R it ablates expression [18], in GLIC it causes a 10-fold decrease in potency [19], while in ELIC it has the opposite effect with a 5-fold decrease in EC_50_ [20]. Our data indicate that this residue can play a role in both expression and function in the GlyR, and these features can be explained by its interactions with nearby residues. The structural data (Figure 7) suggest P30 could form a Pro-π interaction with W94, one of a pair of conserved Trps (the other is W68) located deep in the β sandwich of each subunit ECD, and whose substitution with Ala ablates expression, indicating a critical role of this aromatic pair in protein assembly or membrane targeting [21]. P30 could stabilize W94, an interaction that may be less critical for the integrity of the subunit when protein folding is slower as in oocytes. Noncanonical amino acid substitutions have been used to explore W94, and demonstrated that its size and shape are especially important, not only for expression but also for function, which the authors suggested was via changes to the binding site loops A and D [21]. The current data from P30 extend these findings, showing again that size and shape of a residue in this location alters receptor function. This is particularly evident from the effects of substituting 2MeP and 3MeP, which have opposite effects on the receptor: 2MeP is deleterious, causing a 5-fold increase in EC_50_, while 3MeP results in a gain of function with a ~10-fold decrease in EC_50_. A methyl group can affect the pucker of the Pro ring and might alter the bend in the α1β1 loop, and/or it could alter interactions with local residues such as those in loop D (Figure 6). We propose both of these are likely: previous work has shown Ala substitutions at D70 and Y24 cause similar increases in EC_50_ to 2MeP, providing support for the hypothesis that the methyl group here may prevent interaction of these residues [22,23]. The gain of function with 3MeP could be due to a change in the bend here, and support for this comes from the gain of function caused by Pip and Vah, both of which could alter the tight α1β1 loop turn, but might also be a result of altered interactions perhaps due to an increase in the hydrophobicity in the interior of the ECD; this could help stabilise the critical W94-W68 interaction. Overall the data support the hypothesis that this region of the GlyR ECD plays roles in both assembly and function, and that the interactions of P30 contribute significantly to both these roles.

P96 showed a large change in EC_50_ when substituted with Ala. It is part of loop A and thus an Ala here could directly alter binding interactions and/or the local structure, changing binding affinity. The taurine data, however, show I_max_Tau/I_max_Gly in P96A-containing receptors differs from WT, and thus do not support this hypothesis. The structure (Figure 8) reveals P96 is < 4 Å from W94, and thus could form a stabilizing Pro-π interaction here as described above for P30, but the fact that the Tau data differ from those of P30 suggest this is not its major role. An alternative explanation could involve the reduced hydrogen bonding capacity of Pro, as substitution with Vah, which replicates this, has a WT EC_50_. We note there is a conserved Gln residue (Q66) < 4Å from P96 whose location differs in open and closed receptors; we speculate that if this could hydrogen bond with the residue at position 96, it might impede agonist induced conformational change in this region and hence alter efficient gating. The larger Pro analogues also resulted in some loss of function, indicating size is a factor here, with 3MeP being the most deleterious (~5 fold increase in EC_50_, Table 2). This may be because larger groups here could disrupt interactions in the center of the ECD, with 3MeP the most disruptive as its methyl group could clash with Q66. The importance of this Pro varies in different pLGICs: the equivalent Pro has been shown to be important in 5-HT_3_R where substitution with Ala results in no expression [18], but a similar substitution in GLIC has no effects [20], while in ELIC it causes a decrease in EC_50_ [19]. We speculate that this may be due to different interactions of the Pro with the different residues in the ECD interior.

P146 is the most conserved Pro in the pLGIC family. It is located at the apex of the Cys-loop and has been studied in many pLGIC, nearly all of which are nonfunctional when this Pro is substituted with Ala [18,22,24]. Noncanonical amino acid studies in some of these receptors have shown that the enhanced propensity of Pro to form a *cis* peptide bond is critical for function, consistent with high resolution structures showing the *cis* peptide bond in many—but not all—receptors (see Mosesso et al. for discussion of why they are not all *cis* [19]). Other factors also contribute, and these vary between receptors [20]. In the GlyR these other factors appear less important, as a plot of *cis-trans* preference versus relative EC_50_ from our data shows a strong relationship (Figure 9A), as does our I_max_ Tau data versus % *cis* (Figure 9B), consistent with a *cis* prolyl peptide bond playing a role in the function of the receptor.

However, if a *cis* prolyl peptide bond is so important, then why did we observe functional receptors when P146 was substituted with Ala? One possible explanation is that a *cis* peptide bond is not essential in the GlyR, but we consider it is more likely that the surrounding structure and/or specific local amino acids are such that the *cis* peptide bond can still form with Ala, even if its propensity to form such a bond (0.06%) is much less than Pro (8%) [27]. We note there are many aromatics residues close to P146; aromatic residues adjacent to a Pro assist *cis* peptide bond formation, and these could perhaps help stabilise a non-Pro *cis* bond. We also observe that there are three local Met residues and that interactions between an S-containing amino acid and an aromatic ring are very favourable: a study in 2012 found 33% of structures in the protein database had at least one Met-aromatic motif [28] and have been estimated to contribute 0.5–2 kcal/mol to stability [29]. A speculative hypothesis is that such interactions here could assist in forming a structure that stabilizes a *cis* peptide bond even in the absence of Pro.

P230 is located in the M1 helix. An equivalent Pro is present in all pLGICs and has been shown to be essential in the agonist binding subunit for receptor activity in nAChR and 5-HT_3_R [30,31]. In these proteins, this Pro displays a distinctive phenotype: conventional mutants are nonfunctional, but incorporation of α-hydroxy residues yields robustly expressing functional receptors. Thus of the several unique features of Pro, lack of a backbone hydrogen bond donor is the essential requirement here. The same pattern, however, is not found in all pLGICs; in GLIC this Pro is very sensitive, with only very few close Pro analogues, not including α-hydroxy residues, yielding functional proteins, while in ELIC an Ala does not change function [19]. The Pro in GABAρR lies between these 2 extremes with an Ala substitution yielding functional receptors but with a 7-fold increase in EC_50_ [32] Our data suggest the GlyR is similar to the GABAρR, with a Ala substitution resulting in a 5-fold increase in EC_50_, similar to the EC_50_ change seen with a P230S GlyR mutation identified in a hyperekplexia patient [33]. Extending our knowledge with the substitution of Pro analogues again suggests that it is the reduced hydrogen bonding activity that is necessary for efficient gating, as an α-hydroxy residue restores WT functional characteristics. This would impart flexibility to M1, and data from the GlyR and other pLGICs show that there are indeed differences in the structures of M1 close to this Pro in open and closed states of the receptor (Figure 10). Of note is the location of Q226 in the turn above P230; the interaction of Q226 with R271 in the M2-M3 loop is a key energetic pathway for activating GlyR [33,34], with P230 likely influenced by this interaction. The structure also shows that the closest M2 residue is another Gln (Q266), with some differences in this region of M2 in the open and closed states, which moves away from the axis of the pore when the channel opens; thus, the flexibility of M1 may be needed to accommodate this. This would be accomplished by Pro or analogues with reduced H bonding capacity, with incorporation of 3MeP perhaps destabilizing the M1 helix or the interaction with M2 (the methyl group here would face M2) which could facilitate conformation change.

## 5. Conclusions

In conclusion, we have used functional data from receptors incorporating both canonical and noncanonical amino acids, combined with structural information, to reveal the influence and interactions of four important Pros in the GlyR. The data reveal that for efficient function the Pro in the α1β1 loop is needed to create a turn and to be the correct size and shape to interact with nearby residues; the peptide bond of the Pro in the Cys loop requires the *cis* conformation; and the Pros in loop A and M1 allow efficient function because of their reduced hydrogen bonding capacity. All of these residues are conserved in the pLGIC family and thus our data can be broadly extrapolated to other members of the family. The new information will contribute to a full understanding of the mechanism of action of these critical neuronal proteins.

## Figures and Tables

**Figure 1 membranes-12-01012-f001:**
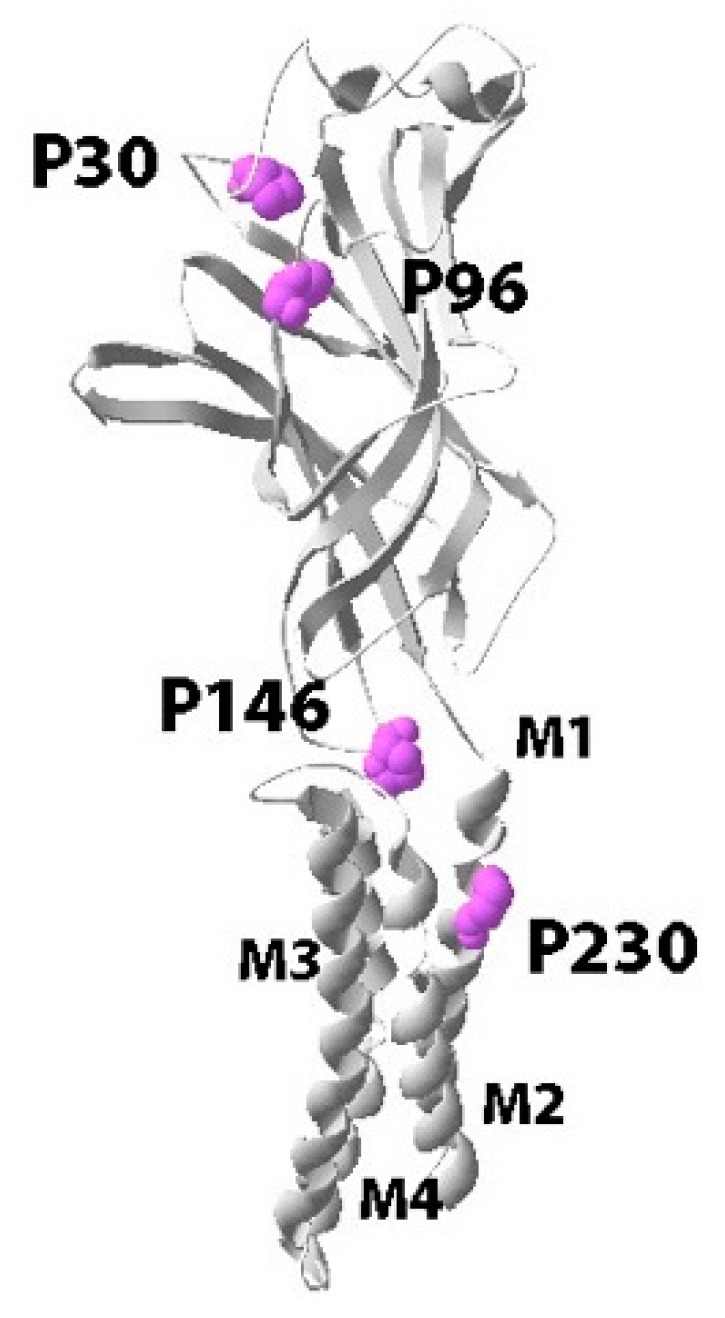
Cartoon of a GlyR α1 subunit showing the location of Pros examined in this study. P30 and P96 are in the ECD, P146 is at the ECD/TM interface, and P230 is on M1 in the TM domain.

**Figure 2 membranes-12-01012-f002:**
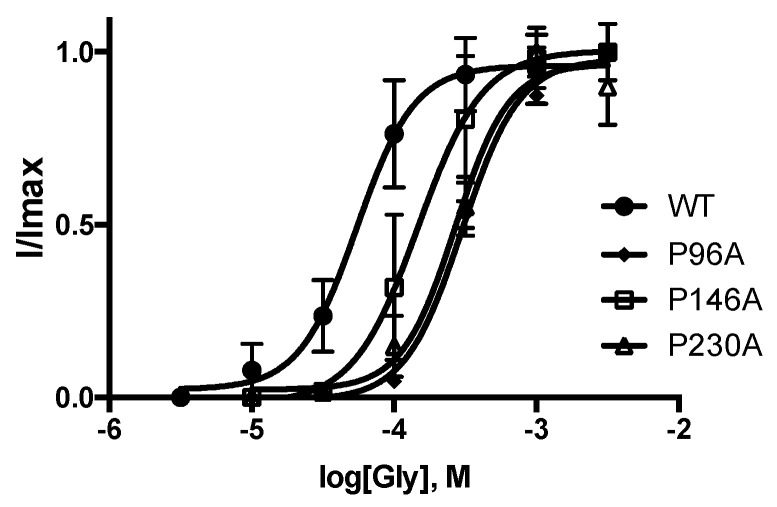
Concentration response curves for WT and mutant receptors derived from maximal responses (see Figure 3 for examples). Data = mean ± SEM, n = 5. Parameters derived from these data are shown in Table 1.

**Figure 3 membranes-12-01012-f003:**
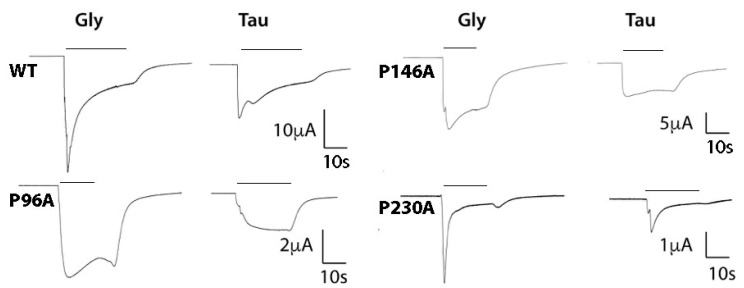
Typical Gly (1 mM) and Tau (10 mM)-induced traces for WT and mutant receptors expressed in oocytes and clamped at −60 mV. Bars indicate agonist application. The different shapes of the responses suggest the mutations may alter receptor kinetics (e.g., desensitization, association and/or dissociation rates).

**Figure 4 membranes-12-01012-f004:**
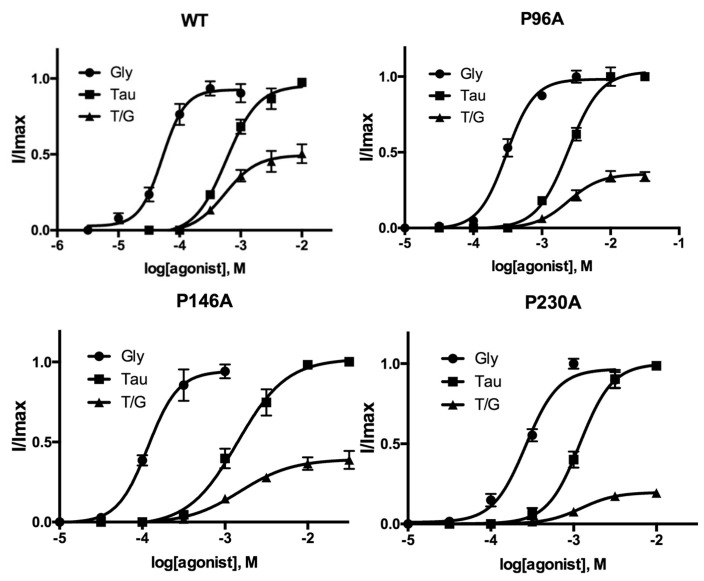
Concentration response curves for WT and mutant receptors T/G = I_Tau_/I _max Gly_. Data = mean ± SEM, n = 5. Parameters derived from these data are shown in Table 1.

**Figure 5 membranes-12-01012-f005:**
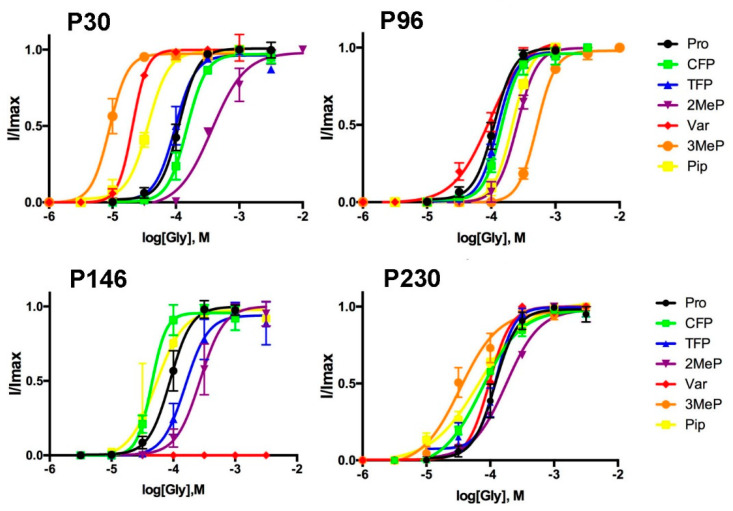
Concentration response curves for WT and mutant receptors substituted with Pro analogues. Data = mean ± SEM, n = 4–6. Parameters derived from these data are shown in Table 2.

**Figure 6 membranes-12-01012-f006:**
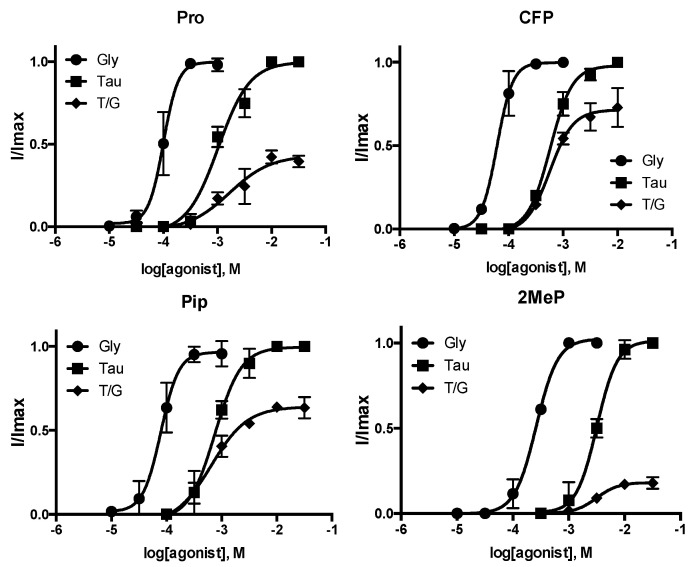
Concentration response curves for GlyR expressed in oocytes where P146 was substituted with Pro or Pro analogues using nonsense suppression. T/G = ITau/I _max_Gly. Parameters derived from these data are shown in Table 3. Data are mean ± SEM, n = 3–4.

**Figure 7 membranes-12-01012-f007:**
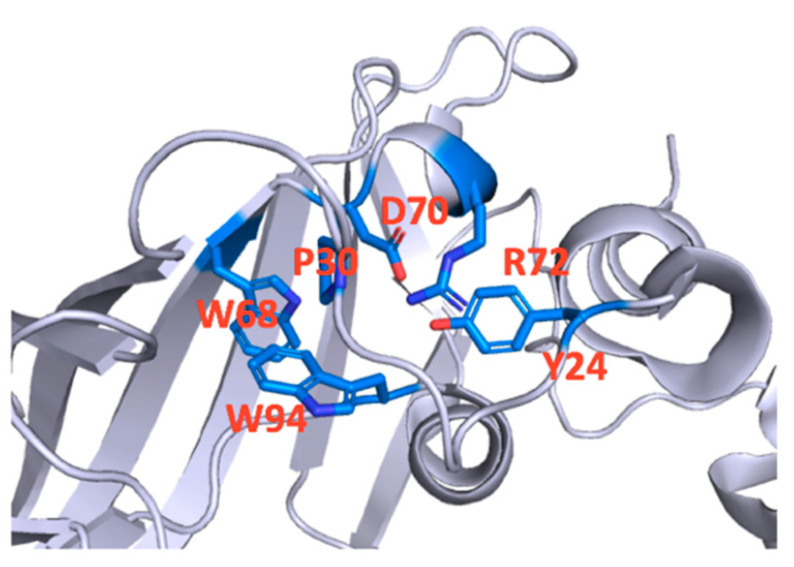
P30 is located at a tight turn in the α1β1 loop and its substitution could alter this turn and/or interactions with or between adjacent residues such as the conserved W68, D70 or R72 in loop D, W94 in loop A or Y24 in the α1β1 loop.

**Figure 8 membranes-12-01012-f008:**
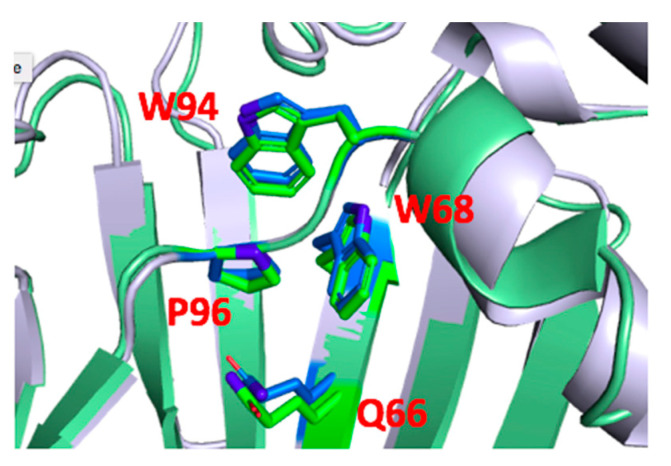
P96 faces the interior of the subunit ECD and its distance from Q66 differs slightly in the open (green) and closed (blue) GlyR structures.

**Figure 9 membranes-12-01012-f009:**
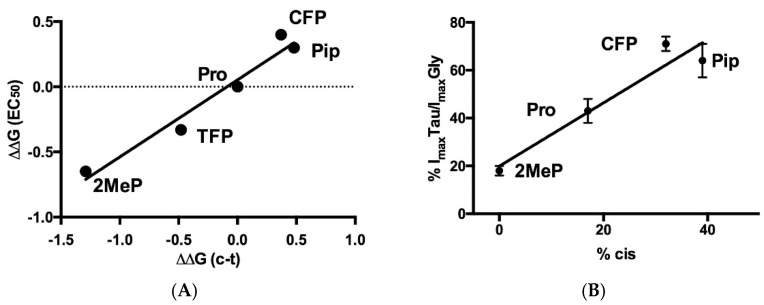
(**A**) Relationship between EC_50_s and *cis-trans* preferences for Pro and analogues at position 146. All values are relative to Pro. R^2^ = 0.96, slope = significantly non zero *p* = 0.004. *Cis-trans* (c-t) values for Pip and CFP from Limapichat et al. [24], for TFP from Pandey et al. [25], and for 2MeP from Kang & Park [26]). (**B**) Relationship between maximal Tau /Gly induced responses (Figure 4) and % *cis* (from Limapichat et al. [24]); R^2^ = 0.92, slope = significantly non zero *p* = 0.04.

**Figure 10 membranes-12-01012-f010:**
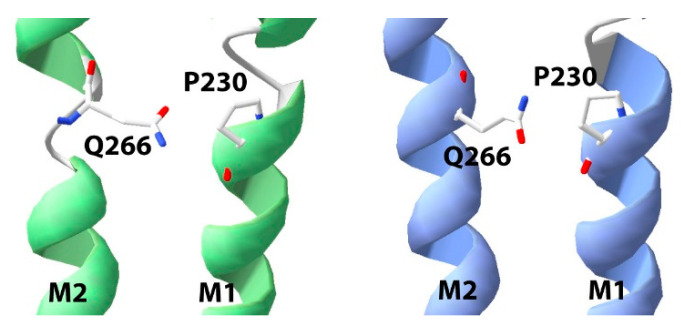
The M1 helix has different structures in the open (green) and closed (blue) states of the GlyR in the region just above P230, and there is also a difference in the adjacent part of M2. The closest M2 residue, Q266, is 2.8Å from P230 in the former and 3.2Å in the latter.

**Table 1 membranes-12-01012-t001:** Parameters derived from WT and mutant GlyR expressed in oocytes.

	Gly			Tau				
	pEC_50_, M	EC_50_ μM	n_H_	pEC_50_, M	EC_50_ μM	n_H_	% I_max_Tau /I_max_Gly	n
**WT**	4.282 ± 0.05	52	2.3 ± 0.5	3.235 ± 0.05	583	1.7 ± 0.3	50 ± 4	5
**P96A**	3.518 ± 0.03 *	304	2.2 ± 0.3	2.612 ± 0.03 *	2440	1.9 ± 0.2	36 ± 2 *	5
**P146A**	3.932 ± 0.05 *	117	2.4 ± 0.7	2.850 ± 0.06 *	1410	1.5 ± 0.3	39 ± 3 *	5
**P230A**	3.579 ± 0.03 *	264	2.2 ± 0.4	2.925 ± 0.03 *	1190	2.2 ± 0.4	20 ± 1 *	5

pEC_50_ = −log EC_50_; n_H_ = Hill coefficient. Data = mean ± SEM, n = 5. * = significantly different to WT, *p* < 0.05.

**Table 2 membranes-12-01012-t002:** Parameters derived from Pro-substituted GlyR expressed in oocytes using nonsense suppression.

Site	Substitution	pEC_50_, M	EC_50,_ μM	n_H_	n
**P30(** **α1β1L)**	Pro	3.948 ± 0.04	113	2.9 ± 1.2	6
	CFP	3.825 ± 0.06	150	2.8 ± 0.7	6
	TFP	4.013 ± 0.05	97	2.7 ± 1.3	5
	Pip	4.439 ± 0.03 *	36	2.7 ± 0.7	4
	2MeP	3.434 ± 0.06 *	368	1.6 ± 0.3	5
	3MeP	5.046 ± 0.05 *	8.9	3.0 ± 2.7	4
	Vah	4.684 ± 0.03 *	21	3.8 ± 0.5	4
**P96 (loopA)**	Pro	3.951 ± 0.04	112	2.7 ± 1.1	5
	CFP	3.840 ± 0.05	145	3.2 ± 0.7	4
	TFP	3.898 ± 0.03	127	2.9 ± 0.7	4
	Pip	3.667 ± 0.02 *	215	2.9 ± 0.3	4
	2MeP	3.596 ± 0.03 *	254	2.8 ± 0.5	4
	3MeP	3.289 ± 0.02 *	514	3.0 ± 0.2	4
	Vah	4.029 ± 0.05	94	1.5 ± 0.3	4
**P146 (CysL)**	Pro	4.052 ± 0.02	89	2.5 ± 0.4	6
	CFP	4.350 ± 0.03 *	45	3.7 ± 0.6	6
	TFP	3.806 ± 0.05 *	156	2.3 ± 0.4	5
	Pip	4.279 ± 0.11 *	53	1.9 ± 0.7	5
	2MeP	3.568 ± 0.04 *	270	2.2 ± 0.5	6
	3MeP	ND			
	Vah	NR			6
**P230 (M1)**	Pro	3.922 ± 0.06	119	2.5 ± 0.9	4
	CFP	4.131 ± 0.04	74	1.4 ± 0.2	4
	TFP	3.904 ± 0.06	125	3.0 ± 1.4	4
	Pip	4.166 ± 0.08	68	1.1 ± 0.2	4
	2MeP	3.751 ± 0.08	177	1.6 ± 0.5	4
	3MeP	4.471 ± 0.08 *	34	1.1 ± 0.4	4
	Vah	4.015 ± 0.02	97	2.6 ± 0.4	4

pEC_50_ = −log EC_50_; n_H_ = Hill coefficient. Data = mean ± SEM. * = significantly different to WT, *p* < 0.05. NR = non-responsive; ND = not determined.

**Table 3 membranes-12-01012-t003:** Parameters derived from P146-substituted GlyR expressed in oocytes using nonsense suppression.

	Gly			Tau				
	pEC_50_, M	EC_50_ μM	n_H_	pEC_50_, M	EC_50_ μM	n_H_	% I_max_Tau /I_max_Gly	n
**Pro**	3.999 ± 0.03	100	3.2 ± 1.6	2.984 ± 0.07	1030	1.5 ± 0.3	43 ± 5	4
**CFP**	4.213 ± 0.04 *	61	3.0 ± 0.5	2.139 ± 0.03 *	579	2.1 ± 0.2	71 ± 3 *	3
**Pip**	4.103 ± 0.05 *	79	2.7 ± 0.9	3.121 ± 0.05 *	757	1.8 ± 0.3	64 ± 7 *	3
**2MeP**	3.582 ± 0.02 *	261	3.6 ± 0.2	2.497 ± 0.04 *	3184	2.4 ± 0.6	18 ± 2 *	3

pEC_50_ = −log EC_50_; n_H_ = Hill coefficient. Data = mean ± SEM. * = significantly different to WT, *p* < 0.05.

## Data Availability

Not applicable.

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
