# Peer review of "Expression of Mutant Glycine Receptors in Xenopus Oocytes Using Canonical and Non-Canonical Amino Acids Reveals Distinct Roles of Conserved Proline Residues"

_membranes, 2022, doi:10.3390/membranes12101012_

Round 1
Reviewer 1 Report
The authors here identify investigated the role of conserved Prolines in the pentameric ligand-gated ion channels (pLGIC). pLGIC are allosteric receptors that mediate rapid electrochemical signal transduction through the opening of the pore upon the binding of ligands. The authors replaced the prolines with canonical (alanine) and noncanonical (proline analogs) amino acids and then carried out functional studies of the mutant channels on Xenopus oocytes using a two-electrode voltage clamp configuration. They found that the Proline in the a1b1 is essential for its interactions with residues around by forming a turn. In addition, the prolines in loops A and M1 are important for normal channel function due to their decreased hydrogen bonding capacity. Since these prolines are conserved in all members of the pLGIC family, the prolines are needed to keep all the channels in the family functional. The data seem convincing, and the conclusion seems appropriate. However, I might have some comments below.
1. Please provide more details in the Figure 3 legend, such as the voltage protocol, and the place where the authors measured and normalized the current.
2. Figure 3, the horizontal legend is needed for WT and P146A. Also, the current trace in WT overlapped with P96A.
3. English editing is needed as some long and complicated sentences are confusing, such as Line 156 and Line 194. Also, the organization of the manuscript needs to check, for example, there are extra spaces between words in some sentences.
4. Lines 295 and 301, missing contexts
5. Figure. 1, please describe the figure in more detail in the legend, also show the names and numbers of the prolines, M1-M4, loops in the figure.
6. Please show a better arrangement of the figures in Figures 6 and 3.
Author Response
The authors here identify investigated the role of conserved Prolines in the pentameric ligand-gated ion channels (pLGIC). pLGIC are allosteric receptors that mediate rapid electrochemical signal transduction through the opening of the pore upon the binding of ligands. The authors replaced the prolines with canonical (alanine) and noncanonical (proline analogs) amino acids and then carried out functional studies of the mutant channels on Xenopus oocytes using a two-electrode voltage clamp configuration. They found that the Proline in the a1b1 is essential for its interactions with residues around by forming a turn. In addition, the prolines in loops A and M1 are important for normal channel function due to their decreased hydrogen bonding capacity. Since these prolines are conserved in all members of the pLGIC family, the prolines are needed to keep all the channels in the family functional. The data seem convincing, and the conclusion seems appropriate. However, I might have some comments below.
- Please provide more details in the Figure 3 legend, such as the voltage protocol, and the place where the authors measured and normalized the current.
We have now provided more detail in the figure 3 and in the figure 1 legend and also added further information in the methods section
- Figure 3, the horizontal legend is needed for WT and P146A. Also, the current trace in WT overlapped with P96A.
Figure 3 has now been revised
- English editing is needed as some long and complicated sentences are confusing, such as Line 156 and Line 194. Also, the organization of the manuscript needs to check, for example, there are extra spaces between words in some sentences.
Long complicated sentences such as those at lines156 and 194 have been modified, and other minor corrections corrected.
- Lines 295 and 301, missing contexts
These have now been added
- Figure. 1, please describe the figure in more detail in the legend, also show the names and numbers of the prolines, M1-M4, loops in the figure.
Fig 1 and its legend have been modified to clarify
- Please show a better arrangement of the figures in Figures 6 and 3.
Figs 3 and 6 have been modified but not rearranged as we feel they adequately demonstrate the data
Reviewer 2 Report
This manuscript described an in-depth study of several Proline amino acids of the GlyR. The experiment is well designed and the paper is well written. It is worthwhile to accept the paper for publication. I have only several minor suggestions or corrections before acception.
1) The name of the amino acids should be written in full at the first appearance, for example, the Pro, Ala and etc. Similarly, in Line 149, "sig diff" should also be written in full. The Tau (chemical) should also be written in full or explained at the first appearance. The "nH" in Table 1, should be explained in the legend or text.
2) In the following positions, probably there are something missing or incorrect in the downloaded pdf.
line 79, human GlyR 1subunit
line 110, the homomeric 3 GlyR
line 111, in the 1 and GlyR
line 267, ’ro's ability
line 404, a Ala to an Ala
3) In figure 3, it is better to indicate when is the agonist applied.
Furthermore, it seems that in Figure 3 the amplitudes and desensitizations are also affected by the mutations, please consider if they are significant, and if true, it might be meaningful to provide a figure for these.
In figure 6, It is not very easy to distinguish some symbols, the lines can be thinner or the symbols can be bigger.
Author Response
This manuscript described an in-depth study of several Proline amino acids of the GlyR. The experiment is well designed and the paper is well written. It is worthwhile to accept the paper for publication. I have only several minor suggestions or corrections before acception.
1) The name of the amino acids should be written in full at the first appearance, for example, the Pro, Ala and etc. Similarly, in Line 149, "sig diff" should also be written in full. The Tau (chemical) should also be written in full or explained at the first appearance. The "nH" in Table 1, should be explained in the legend or text.
These corrections have now been made
2) In the following positions, probably there are something missing or incorrect in the downloaded pdf.
line 79, human GlyR 1subunit
line 110, the homomeric 3 GlyR
line 111, in the 1 and GlyR
line 267, ’ro's ability
line 404, a Ala to an Ala
Yes indeed. Some of these were lost in the conversion but we have now corrected them
3) In figure 3, it is better to indicate when is the agonist applied.
This has now been added to modified Fig 3
4) Furthermore, it seems that in Figure 3 the amplitudes and desensitizations are also affected by the mutations, please consider if they are significant, and if true, it might be meaningful to provide a figure for these.
Yes it is possible that desensitization and/or dissociation rates are affected by the mutations, and we already comment in the manuscript that “the change in EC50 is more likely to be caused by a change in efficacy ….consistent with the ….change in kinetics of the responses” and refer to fig 3 where the shape of the responses differ. We have now also clarified this in the figure legend. However more detailed studies are needed to confirm this which are beyond the scope of this study.
Amplitudes again could be affected, but as these are also at least partly dependent on the individual oocyte, which of course varies for each mutant, we cannot make any definitive statements from our data.
5) In figure 6, It is not very easy to distinguish some symbols, the lines can be thinner or the symbols can be bigger.
We have modified Fig 6
Round 2
Reviewer 1 Report
The authors have addressed all my comments.